# RL Simplex: Bringing Computational Efficiency in Linear Programming via Reinforcement Learning

## Abstract

In the simplex method, the selection of variables during the pivot operation in each iteration significantly impacts the overall computational process. The primary objective of this study is to provide explicit guidance for the selection of pivot variables, particularly when multiple candidate variables for pivoting are available, through the application of reinforcement learning techniques. We illustrate our approach, termed RL Simplex, to the Euclidean Traveling Salesman Problem (TSP) with varying city counts, substantially reducing the number of iterations. Our experimental findings demonstrate the practical feasibility and successful integration of reinforcement learning with the simplex method, surpassing the performance of established solver software packages such as Gurobi and SciPy.

## 1 Introduction

The simplex method is a widely applied mathematical technique for solving linear programming (LP) problems. It operates through a systematic process of moving from one vertex, referred to as a basic feasible solution, to an adjacent one along the edges of a convex polytope. This polytope is essentially a geometric shape with flat sides that is defined by the constraints imposed by the problem. This movement is accomplished through a series of mathematical operations known as pivoting. During each pivot operation, the simplex method selects an entering variable and a leaving variable. The goal is to enhance the value of the objective function while adhering to the problem's constraints. These stepwise adjustments and their associated data, including variable values and objective function details, are documented within a data structure called a *tableau*. This tableau serves as a comprehensive record of the current iteration. The process continues iteratively until no further improvement is attainable, ultimately leading to the determination of the optimal solution (Dantzig, 1963).

The pivot rule is a fundamental component of the simplex method that governs how the algorithm selects variables to enter or leave the basis during each iteration in its search for the optimal solution. Essentially, it functions as a decision-making mechanism, guiding the algorithm's movements within the solution space. The choice of pivot rule significantly influences the efficiency and convergence of the algorithm, as it dictates the sequence of explored solutions. Several pivot rules are available, each with its unique strategy for variable selection. Prominent examples include Dantzig's rule (Dantzig, 1963), Steep Descent rule (Forrest & Goldfarb, 1992), and Bland's rule (Bland, 1977).

In recent years, attention has turned to the application of reinforcement learning to various optimization problems (Tang et al., 2020; Fawzi et al., 2022; Bello et al., 2016; He et al., 2014; Kool et al., 2018; Alvarez et al., 2017). Specifically, innovative solutions have been proposed, particularly in addressing computational challenges related to the simplex method, there has been a growing recognition of the potential benefits of machine learning methods for improving the simplex method's performance. For instance, (Adham et al., 2021) employs neural networks and boosted decision trees to predict appropriate pivot rules for medium-scale linear programming instances. However, this decision-making approach tends to be overly singular in its pivot selection, lacks consideration for dynamic scenarios, and relies on pre-training and labeling for a specific range of linear programming problems. To improve pivot rules, reinforcement learning-based approaches (e.g. Suriyanarayana et al. (2022)) were introduced, aiming to reduce the number of iterations (e.g.,

by choosing between Dantzig and Steep Descent rules based on the current state). Nevertheless, this method was only evaluated on extremely small instances during the testing phase (i.e., TSP with 5 cities). Additionally, during the training process, the method exhibits a lack of balance in action exploration, often pursuing high-reward actions at the expense of potentially overlooked benefits from another action. Similarly based on reinforcement learning, another approach involves Monte Carlo Tree Search (MCTS) for simplex method rules (Li et al., 2022), aiming to find shorter paths and minimize the required number of iterations. While this method has achieved some success in discovering the shortest paths for the simplex method, its practicality may be affected when dealing with entirely new instances.

In this paper, we aim to improve the simplex pivoting rules using reinforcement learning replacing the most commonly used "largest pivot rule" in the simplex method. The proposed pivoting rule is established based on the exploration and reward mechanisms inherent in reinforcement learning. Our work is a proof of concept and represents a step forward in optimizing computational efficiency and leveraging machine learning. Similar to relevant literature such as Suriyanarayana et al. (2022) we assess the performance of our proposed approach by solving the Linear Programming (LP) relaxation of the Euclidean TSP, with varying city counts and distance intervals, utilizing both learning and acting phases. To provide a fair judgment of the performance, we compare our own implementation of LP solver, and well-established solvers such as Gurobi, CPLEX, and SciPy regarding the number of iterations required to find optimal solutions for LP relaxation of TSP instances. Note that these solvers do not provide an option to change pivoting rules, therefore, to investigate pivoting rules, one needs to implement an LP solver. The results indicate that the proposed Reinforcement Learning (RL) simplex algorithm performs well when dealing with a high number of cities and smaller distance intervals, outperforming established solvers such as Gurobi (Gurobi Optimization, 2015) and SciPy (Community, 2012). Regarding the comparison within the literature, to the best of our knowledge, we solve the largest instances of the TSPs solved using RL-based simplex methods.

In summary, our main contributions are as follows:

- Introducing a novel algorithm that leverages reinforcement learning to enhance the largest pivot coefficient rule in the simplex method.
- Balancing exploration and exploitation in the reinforcement learning process by integrating the UCB (Upper Confidence Bound) algorithm.
- Demonstrating, through numerical experiments, the seamless integration of reinforcement learning and the simplex method, showcasing the potential for significant performance improvements.
- Pioneering the optimization of computational efficiency by combining machine learning with exact algorithms, resulting in reduced iteration requirements while maintaining optimal solutions for the Euclidean TSP instances. Acknowledging that, despite these advancements, scalability challenges persist in addressing complex real-world scenarios involving a large number of cities.

While our algorithm represents a significant step forward since it tackles larger instances of the Euclidean TSP than previously reported in related literature using RL-based simplex methods, we recognize certain limitations in addressing complex, large-scale real-world scenarios. Our research, being at the forefront of this emerging field, opens the door for future advancements and refinement in this area.

## 2 BACKGROUND: MODEL FORMULATION

### 2.1 SIMPLEX METHOD

The simplex method is a widely used method for solving programming problems, which can be expressed in standard form as follows:

$$\text{Minimize } c^T x$$
$$\text{subject to } Ax \leq b, x \geq 0$$

where $c$ is the coefficient vector of the objective function; $A$ is the coefficient matrix for the constraints; $b$ is the vector of constants on the right-hand side of the constraints; and $x$ is the vector of decision variables to be determined using the optimization. The decision variables can be continuous or discrete.

The simplex method starts with an initial feasible solution and iteratively improves it by moving along the edges of the feasible region to find the optimal solution. At each step, the algorithm selects a pivot element and performs a pivot operation to change the basis of the solution. The pivot operation involves selecting a pivot row and a pivot column. The pivot row is chosen based on the minimum ratio test, which ensures that the entering variable does not violate its constraint. The pivot column is chosen based on the most negative coefficient in the objective function. The selected pivot element is used to perform row operations to transform the pivot column into the new basic column. This adjusts the current solution and moves it closer to the optimal solution. The algorithm continues these pivot steps until no further improvement can be made, at which point the optimal solution is found (Hillier, 2001). Subsequently, some other pivot rules were introduced, but this iterative solving approach remains unchanged.

## 2.2 TRAVELING SALESMAN PROBLEM FORMULATION AND PROBLEM DESCRIPTION

The Traveling Salesman Problem is a classic discrete optimization problem, briefly described as the experimental background for our algorithm. The TSP involves finding the shortest possible route that visits a set of cities exactly once and returns to the starting city. It is fundamental in combinatorial optimization and has applications in logistics, transportation, and circuit design (Cook et al., 2011).

In mathematical representation, the TSP can be formulated as follows:

$$\text{Minimize} \quad \sum_{i=1}^{n} \sum_{j=1, j \neq i}^{n} c_{ij} x_{ij} \tag{1a}$$

subject to the following constraints:

Arrival constraint:

$$\sum_{j=1, j \neq i}^{n} x_{ij} = 1, \quad \forall i \in \{1, 2, \ldots, n\} \tag{1b}$$

Departure constraint:

$$\sum_{i=1, i \neq j}^{n} x_{ij} = 1, \quad \forall j \in \{1, 2, \ldots, n\} \tag{1c}$$

Subtour elimination constraint:

$$u_i - u_j + n \cdot x_{ij} \leq n - 1, \quad \forall i \in \{2, 3, \ldots, n\}, \forall j \in \{2, 3, \ldots, n\}, i \neq j \tag{1d}$$

where:

- $n$: The number of cities to be visited in the TSP, where $n \geq 2$. It represents the total count of cities on the tour.
- $c_{ij}$: The cost or distance associated with traveling from city $i$ to city $j$, where $i, j = 1, 2, \ldots, n$. This parameter defines the distance or cost between each pair of cities on the tour. It is used in the objective function to calculate the total tour cost.
- $x_{ij}$: A binary decision variable indicating whether the salesman takes the direct route from city $i$ to city $j$.
- $u_i$: An integer variable representing the position of city $i$ in the tour. It is constrained between 2 and $n$ ($2 \leq u_i \leq n$). These variables establish the order in which cities are visited in a tour i.e., $u_i$ indicates the position of city $i$ in the sequence of a tour.

Consider the setting in which the cost function associated with the TSP varies due to changes in transportation costs daily. This brings up the question of how we can learn from the TSP instances

solved in the past to enhance the computational efficiency of solving the TSP instance which we are facing today. In summary, we are interested in learning within an environment where the coefficients of the objective function 1a are subject to change, while the constraints remain consistent. In this paper, we develop an approach which leverages historical LP instances with unchanged constraints, effectively learning and adapting to the evolving objective function to find optimal solutions efficiently. Note that our focus in this paper is on solving the LP relaxation of the TSP, and that is why we have dropped the integrality constraints of TSP in the formulation 1a-1d.

# 3 REINFORCEMENT LEARNING FOR THE SIMPLEX METHOD

## 3.1 REINFORCEMENT LEARNING APPROACH

In this section, we introduce the methodological approach for the integration of reinforcement learning within the simplex method. Specifically, we expand upon the state representation, action space, and reward function which are the primary components of an RL model. Furthermore, we explain how to address the well-known dilemma of exploration vs. exploitation in RL models.

**State Representation:** In the simplex method, the tableau are updated with each iteration, including comprehensive information pertaining to the current iteration. The algorithm defines a set of all possible simplex tableau as the state representation set S within the reinforcement learning model. The variable $s_t$ is used to denote the state at iteration $t$. Specifically, the tableau includes entries corresponding to the distances $c_{ij}$ between cities, binary variables representing tour edges, position variables $u_i$ that enforce subtour elimination constraints, and slack variables.

**Action Space:** The action space is defined as the set of largest non-basic coefficients corresponding to the variables of the objective function.

**Reward Function Definitions:** In iteration $t$, the reward function $R(s_t, a_t)$ can be defined as a piece-wise function:

$$R(s_t, a_t) = \begin{cases} -k \cdot t, & \text{if } a_t \text{ is the largest non-basic variable in iteration } t, \text{ where } k \in [0, 1] \\ 2000, & \text{if all candidate non-basic variables are negative in state } s_{t+1}. \end{cases} \quad (2)$$

Choosing different actions directly impacts the number of iterations of the simplex method. Unlike common reward functions, whenever an action is chosen, a negative reward of $-k \cdot t$ is received, where the discount factor $k < 1$ is introduced in order to prevent the accumulation of significant negative rewards caused by excessive iteration counts. Conversely, when $s_{t+1}$ satisfies the termination condition of the simplex method in the final round, the action chosen in $s_t$ is rewarded significantly.

**The Exploration vs. Exploitation Dilemma in Reinforcement Learning:** In contrast to the classical simplex approach that uses the Largest Coefficient Rule, the proposed approach, the RL Simplex algorithm, places greater emphasis on scenarios where multiple candidates contest for the role of the largest pivot. Specifically, the exploration criterion within the approach presented in this paper has the capability to integrate an adjustable scaling factor, drawing from the principles of the Upper Confidence Bounds (UCB) method (Browne et al., 2012). By employing this approach, the UCB algorithm prevents from persistently selecting a singular action, thereby neglecting the potential global rewards associated with other actions. This approach reduces the uncertainty that arises from probabilistic random exploration, a common occurrence in the $\varepsilon$-greedy policy. The UCB algorithm is expressed as follows: Let $N_t(a)$ be the number of times action $a$ has been selected up to iteration $t$, and let $Q_t(a)$ be the estimated value of action $a$ up to iteration $t$. Then, the UCB value $UCB_t(a)$ for action $a$ at time $t$ is given by:

$$UCB_t(a) = Q_t(a) + c \cdot \sqrt{\frac{\log(t)}{N_t(a)}} \quad (3)$$

Here, $c$ is a hyperparameter that controls the level of exploration, $\log(t)$ is the natural logarithm of $t$, the number of time steps elapsed, and $N_t(a)$ represents the number of times action $a$ has been

selected up to time step $t$. It's a count of how often a specific action has been chosen by the agent.

**Q-Value update:** In the context of Q-Learning, the process of refining Q-values is augmented by the incorporation of the UCB strategy, enhancing the algorithm's exploration-exploitation balance. In this dual mechanism, the Q-value $Q(s_t, a_t)$ for a given state $s$ and action $a$ is iteratively adjusted by a composite factor. This factor comprises two components: the immediate reward $r$ obtained from action $a$ in state $s$, and the expected maximum cumulative reward associated with the subsequent state $s_{t+1}$ and its optimal action $a_{t+1}$, weighed by a learning rate $\alpha$ and a discount factor $\gamma$.

Mathematically, the updated Q-value is governed:

$$Q(s_t, a_t) \leftarrow Q(s_t, a_t) + \alpha \cdot \left( r + \gamma \cdot \max_{a_{t+1}} Q(s_{t+1}, a_{t+1}) - Q(s_t, a_t) \right) \tag{4}$$

However, in contrast to the classical Q-Learning scheme, the addition of UCB introduces a controllable scaling factor $\text{ucb}_t(a)$ rooted in the Upper Confidence Bounds method equation 4. This augmented Q-value update mechanism is expressed as:

$$Q(s_t, a_t) \leftarrow Q(s_t, a_t) + \alpha \cdot \left( r + \gamma \cdot \max_{a_{t+1}} Q(s_{t+1}, a_{t+1}) - Q(s_t, a_t) \right) + \text{ucb}_t(a) \tag{5}$$

By seamlessly integrating Q-Learning with the UCB strategy, the algorithm adeptly balances the exploitation of the current pivot information and the exploration of new possibilities, ultimately leading to a more thorough exploration of potential alternative solutions. This synergy enhances the algorithm's learning and decision-making capabilities.

**Novel RL Simplex Algorithm for TSP:** For each instance of the TSP, a number of episodes are required to be executed. The conclusion of an episode signifies the completion of a full cycle through the simplex method, as described in Algorithm 1. Notably, each episode encapsulates a full cycle of the simplex method, and the outcomes of each algorithmic iteration are meticulously recorded within the Q-table.

A trained Q-table will store the $c_{ij}$ values for each TSP instance across different iterations $s_t$, along with their corresponding acquired Q-values. This will serve as a point of reference and guidance for subsequent occurrences of similar scenarios.

---

**Algorithm 1:** RL Simplex: Q-learning Update

**Input** : The Euclidean TSP instance with parameter $c_{ij}$, $A$, $b$, max episodes $T$
**Output:** Updated Q-table

1 **for** *episode $\leftarrow 0$ to $T$* **do**
2      Initialize current state $S_t$ and tableau using the Euclidean TSP instance parameters $c_{ij}$, $A$, and $b$;
3      Initialize previous state $s_{t-1}$ as $-1$;
4      **if** *episode $= 0$* **then**
5          Choose the largest coefficient from the initial Euclidean TSP instance $c_{ij}$ as the action;
6      **while** *not all candidate non-basic variables are negative in state* **do**
7          Choose the action with the largest Q value from the Q table;
8          Observe the next state $s_{t+1}$ and reward;
9          **if** *not all candidate non-basic variables are negative in $s_{t+1}$* **then**
10              Calculate $q\_target$ based on reward and the maximum Q value of the next state;
11              Update the Q value in the Q table;
12              Record the previous action $a_{t-1}$;
13              $s_t \leftarrow s_{t+1}$;
14              Perform the simplex method on the tableau;
15          **else**
16              Calculate $q\_target$ based on final reward $R$; Update the Q value for the previous state and the previously chosen action;
17 Return the updated Q-table;

---

## 4    EXPERIMENTS

This section assesses the performance of the RL Simplex algorithm applied to Euclidean TSPs under various testing conditions. Specifically, experiments were conducted with different numbers of cities, varying distances between cities, and varying degrees of reinforcement learning. The performance of the simplex method was compared to the simplex method without any learning process using several mainstream solvers commonly used to solve TSPs.

### 4.1    EXPERIMENTAL PROCEDURE: LEARNING AND ACTING

The experimental process is divided into two main phases: Learning and Acting displayed in Figure1 and described below.

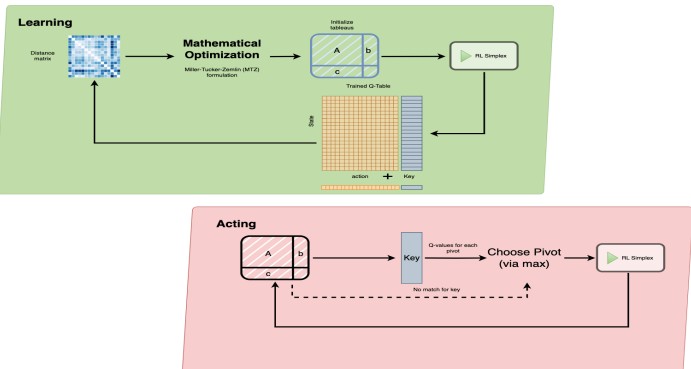

Figure 1: Overview of Experimental Procedures in RL Simplex

**Learning Phase:** During the learning phase, the primary objective is to accumulate experience for the RL Agent through a number of instances. For each distinct TSP instance, we transform the problem into an LP based on the distance matrix between cities, ensuring consistent constraints. This consistency in the action space is maintained when applying reinforcement learning to TSP instances with the same number of cities.

The simplex method is not a one-step process; in the execution of the RL Simplex algorithm, each iteration of the simplex method generates a new LP problem or state denoted as $S$. This implies that the Q-table requires updates based on the action chosen in each iteration. Moreover, the Q-table used in the RL Simplex algorithm deviates from the conventional Q-table. In the final column of the Q-table, we employ the coefficients $c_{ij}$ associated with each state as *Key* indices. This approach allows us to use these coefficients as references for providing feedback on the rewards associated with previous training actions when encountering states in subsequent training and testing phases. To minimize the additional computational costs associated with querying keys, the Q-table is updated in accordance with the iteration order of each instance. The same instance can be tackled repeatedly until a predetermined *Episode* threshold is reached, allowing for the exploration of diverse candidate actions. Moreover, only the solution with the minimum number of iterations is preserved.

**Acting Phase:** During the Acting Phase, each new LP problem generated by the RL Simplex algorithm requires searching the trained Q-table using $c_{ij}$ as an index. However, RL Simplex states serve not only to record the sequence but also include information from the current iteration in the tableau, including city distances $c_{ij}$. Therefore, we need to separately record this new information and name it "key". If a matching key is found in the Q-table, the RL Simplex algorithm selects the corresponding pivot or action $a$ as the entering variable based on the maximum Q-value. Conversely, if no *Key* match is found, the RL Simplex algorithm follows the largest coefficient rule to resolve the current iteration. This process is repeated until the simplex method terminates and the optimal solution is found.

## 4.2 EXPERIMENT DESIGN

**LP instances for Learning and Acting Phase:** For the experiments used to evaluate the performance of the simplex and RL Simplex algorithm, the datasets consisted of random instances with varying numbers of cities and city distance intervals, We generated 1000 instances for the learning phase and 200 for the acting phase. Due to the iterative nature of the simplex method, each iteration can be considered as solving a new programming problem, thus encountering a significantly larger number of instances during the learning phase compared to the initial training dataset. As we selected the Euclidean TSP to illustrate the RL Simplex, instances with the same number of cities share identical constraints ( 1b- 1d). Consequently, each instance has a size of $1 \times n$, encompassing the distance coefficients $c_{ij}$, position variables $u_i$, and slack variables as defined in Section 2.2.

**Comparing Solvers and Pivot Rules:** When employing the simplex method as our chosen solving approach, we opted to compare to the SciPy linprog solver (Community, 2012) and Gurobi (Gurobi Optimization, 2015), both of which are widely recognized as established solvers in the domain of LP problems. In order to account for the influence of problem instance size on these solvers, we also conducted a comparative analysis of our approach against the HiGHS solver. HiGHS is specifically engineered for high-performance serial and parallel computing, tailored to address large-scale sparse LP, mixed-integer programming, and related problem instances (Huangfu & Hall, 2018). The primary purpose of employing the Bland rule (Bland, 1977) as the upper bound criterion is because the Bland rule is utilized to prevent the simplex method from entering into cycling behavior, thus guaranteeing termination within a finite time frame. Consequently, the Bland rule often necessitates a higher number of iterations to reach a solution. Furthermore, to validate the effectiveness of our algorithm during the learning process, we conducted a performance comparison with the RL algorithm in the absence of a learning phase, which we refer to as RL Simplex (Base) in the rest of this paper.

## 4.3 CRITERIA IMPACTING COMPUTATIONAL PERFORMANCE

**Varying Distance Intervals between Cities:** The simplex method is highly sensitive to the coefficients of the objective function, particularly under the Largest Coefficient Rule (Dantzig, 1963). To minimize the impact of other factors in subsequent experiments with varying numbers of cities, we conducted experiments with different distance intervals as the background for the TSP with 5 cities. We compared three different city distance interval scenarios, which are the ranges of $c_{ij}$: 1–5, 1–15, and 1–20, as shown in Table 4.3. The results indicate that the RL Simplex with the Largest Coefficient Rule requires more iterations to converge as city distances decrease. Conversely, the other comparison methods exhibit the opposite trend, which aligns with our hypothesis that the convergence rate of the Largest Coefficient Rule slows when city interval distances have minimal differences or are equal.

Table 1: *City Range Analysis - Average Number of Iterations for Different Solvers and Rules.*

| City Range | Solvers and Rules (#iterations[1]) | | | | | |
|---|---|---|---|---|---|---|
| | SciPy solver (Simplex) | Bland rule | HiGHS solver | Gurobi (Simplex) | RL Simplex (Base) | RL Simplex (Learned) |
| 1-20 | 60.9 | 71.1 | 34.4 | 20.0 | 15.6 | 13.9 |
| 1-15 | 61.5 | 70.6 | 33.8 | 18.5 | 15.7 | 14.1 |
| 1-5 | 57.7 | 65.9 | 32.1 | 18.3 | 17.0 | 13.8 |

[1] The average number of required iterations across 200 test sets.

**Varying the City Count:** When evaluating the performance of the RL Simplex algorithm, it is essential to consider the scale of the problem, specifically, the number of cities, and how it affects the algorithm. The dimensions of the Q-table used in the algorithm vary with the number of cities. An increase in the number of cities not only increases the available actions and states but also introduces more constraints. TSP instances for different city counts necessitated varying numbers of variables, as shown in Table 2.

In our experiments, the primary focus was to assess the algorithm's scalability and efficiency. The most notable outcome is that, as the number of cities increases, the average number of iterations

required by the algorithm significantly rises. For example, when expanding from 5 cities to 7 cities, the results for both SciPyy (Community, 2012) and the Gurobi (Gurobi Optimization, 2015) commercial solver doubled. This is not an encouraging sign, as it suggests that in larger-scale problems, such as with 50 cities, these solvers might require a substantial number of iterations to reach an optimal solution, or even fail to find one within the defined limitation.

It's worth noting that for the 50-city test, the upper limit for the number of iterations was set at $5100$. In contrast, the results obtained by the RL Simplex algorithm are much more promising, demonstrating its outstanding scalability when handling larger-scale problems compared to other simplex solvers.

Moreover, in terms of efficiency, the untrained RL Simplex algorithm outperforms other methods in all tests, except for HiGHS (Huangfu & Hall, 2018). This performance difference becomes more noticeable with increasing problem complexity. When compared to HiGHS, a solver designed for complex and large-scale LPs, the basic version of our algorithm gradually lags behind. The reason for introducing HiGHS is to provide a better comparison for the changes in our RL Simplex algorithm before and after training. Considering the results in Table 2, it is evident that with limited training, RL Simplex can approach or even surpass the results obtained with HiGHS, further demonstrating the efficiency and practicality of our algorithm when compared to non-Simplex algorithms.

Two primary factors contribute to this phenomenon. 1) Untrained RL Simplex algorithm exhibit significant uncertainty in pivot selection. This suggests that relying on the untrained RL Simplex algorithm as a baseline presents ample opportunities for improving the learning process. For example, in a test case involving 50 cities, the variation in iteration counts due to different pivot selections in a single instance can be as high as $4850+$. 2) As the complexity of the problem itself increases, the number of required iterations also rises, which is what we hope to see in the training process because it provides more training data.

Table 2: *City Count Analysis - Average Number of Iterations for Different Solvers and Rules.*

| City Count (Variables[1]) | Solvers and Rules (#iterations) | | | | | |
|---|---|---|---|---|---|---|
| | SciPy solver (Simplex) | Bland rule | HiGHS solver | Gurobi (Simplex) | RL Simplex (Base) | RL Simplex (Learned) |
| **5 cities** (24/12/36) | 57.7 | 65.9 | 32.1 | 18.3 | 17.0 | 13.8 |
| **7 cities** (48/30/78) | 203.9 | 202.9 | 21.1 | 39.0 | 22.1 | 19.8 |
| **10 cities** (99/72/171) | 665.6 | 613.9 | 35.7 | 94.0 | 55.1 | 33.9 |
| **15 cities** (224/182/406) | 3305.2 | 1389.3 | 55.9 | 349.4 | 134.2 | 49.5 |
| **50 cities** (2499/2352/4851) | 5100+[2] | 5100+[2] | 317.1 | 4979.2 | 3858.4 | 387.3 |

[1]Variables represent the counts of Decision Variables/Slack Variables/Total Variables. [2] The method used could not provide an optimal solution within 5100 iterations. [3] The best outcomes are represented by the color red, while suboptimal results are indicated by the color blue.

## 5  DISCUSSION

**The Influence of Various Factors on Outcome:** In the pivot selection phase, the number of available candidates can also influence the outcomes of the RL Simplex algorithm. To investigate this, we conducted a statistical analysis of the occurrences of the same largest distance in the training and test sets for different quantities of cities used in the "Varying the City Count" analysis in Section 4.3. As shown in Table 3, when the number of cities increases, the average occurrence of the same largest distance also increases. This underscores the need for increased investment in the learning phase, including longer episodes and larger training datasets, to help reduce the number of iterations in the acting phase. Additionally, the presence of multiple available largest pivot candidates provides the RL Simplex with more opportunities to explore paths with fewer iterations, a pivotal factor that distinguishes it from the basic simplex method.

Table 3: *Average Occurrence Count vs. City Count (Test and Train).*

| City Count | 50 | 15 | 10 | 7 | 5 |
|---|---|---|---|---|---|
| Average Occurrence Count (Test) | 77.9 | 10.1 | 7.5 | 5.7 | 4.3 |
| Average Occurrence Count (Train) | 79.4 | 10.8 | 7.4 | 5.6 | 4.4 |

**Impact of Query Time on Total Algorithm Execution Time:** In this section, we investigate the trade-off between the reduction in the number of iterations and the additional computational cost incurred by querying keys from the pre-trained Q-table. For this experiment, we randomly selected instances from different city ranges and individually tested the RL Simplex algorithm, assuming that these instances were previously learned during the training phase to ensure access to learned information from the pre-trained Q-table. As shown in Figure 2, the proportion of computational time spent on key queries relative to the overall runtime consistently decreases as the city range expands. Particularly, for larger-scale instances, the Q-table query time accounts for only 2% of the total time which demonstrates the advantage of the RL pivoting rule. In other words, the computational cost for reducing the number of iterations from 3000+ in all solvers (except HiGHS) to 239 in our approach is insignificant for large instances.

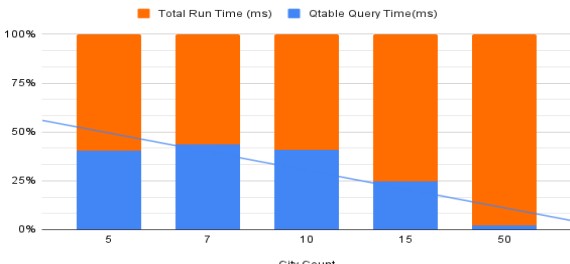

Figure 2: Query Time vs. City Count

**Practical Analysis and Challenges:** The practicality of algorithms is always a primary focus of algorithm development studies. To this end, we conducted experiments with varying numbers of cities to demonstrate the scalability of the RL Simplex algorithm. However, we are mindful that these experiments only scratch the surface of numerical exploration. To efficiently solve larger instances of the TSP, algorithms specifically developed for the TSP are required. Particularly, valid inequalities such as comb inequalities and various types of multistar inequalities have been used to improve the presentation of the TSP's convex hull. For more details on the computational efficiency of the TSP solution method, the reader is referred to Jens Lysgaard (2004) and Cook et al. (2011). Many of these valid inequalities are compatible with the application of our approach, i.e., for a given problem size, the constraints remain unchanged. Consequently, the RL Simplex algorithm is not limited to the plain structure of the TSP and can further improve the scalability.

## 6 CONCLUSION AND FUTURE WORK

Without the integration of machine learning, the pivot selection process in the simplex method primarily relied on mechanistic rules, especially when dealing with multiple available pivot candidates, clear guidance was lacking. However, by incorporating reinforcement learning into the simplex method, we have introduced a more effective pivot selection solution. In the context of the LP relaxation of TSP, covering various city counts and distance intervals, our proposed RL Simplex significantly reduces the number of iterations compared to other mainstream solvers. Certainly, for the TSP, it continues to pose significant challenges and is categorized as an NP-Hard problem (Papadimitriou, 1977). Looking ahead, we plan to extend the application of this algorithm to general linear programming problems and integrate it with neural networks and reinforcement learning to address even more extensive training and testing scenarios. Moreover, we intend to incorporate valid inequalities for the TSP and apply our RL Simplex approach to solve large instances. Additionally, RL-based methods can be applied to separation algorithms to identify violated valid inequalities of the TSP.

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
