# OpenReview forum: "RL Simplex: Bringing Computational Efficiency in Linear Programming via Reinforcement Learning"
_ICLR.cc/2024/Conference — Submitted to ICLR 2024_

### Official Review · Reviewer_PpN6 · 2023-10-16

**Soundness:** 2 fair
**Presentation:** 3 good
**Contribution:** 2 fair
**Rating:** 3
**Confidence:** 5

**Summary:**

This paper propose a new approach, named RL Simplex, to accelerate the simplex iteration in Euclidean Traveling Salesman Problem (TSP). Experiments show the practical feasibility and successful integration of reinforcement learning with the simplex method. The authors claim that their approach outperforms Gurobi and SciPy in terms of the number of iterations.

**Strengths:**

1. Clear writing, the paper is clearly structured and easy to go through flow.
2. The proposed approach is technically sound. The employment of RL in this task is technically sound.

**Weaknesses:**

1. Unclear motivation. Generally, incorporating ML models to the Simplex task is intractable, as simplex in modern solvers is extremely fast (usually faster than $1ms$ for one iteration), while ML models usually require $10$x or even $100$x more time. Simplex iteration usually execute for thousands of times or even more in real-world applications, making the additional cost unacceptable. Previous research [1] based on MCTS claims that their approach can provide the best pivot labels for all kinds of supervised learning methods, but what is the motivation for this paper?
2. Toy applications. LP simplex is widely used in modern solvers for general LP and MILP problems. However, this paper only focus on the TSP problem (with very small size), making this study impractical for real-world applications.
3. Lack of comparative baselines. both [2] and [3] propose similar approaches in this task, what is the comparison between this approach and them? If time is not taken into consideration, then maybe the non-data-driven "strong branching" policy proposed in [2] can outperform some data-driven policies.
4. Reward design is too empirical. The reward design seems to be totally empirical. However, in RL, designing rewards in this way can sometimes result in unexpected agent behaviors. Maybe a reward that completely proportional to the number of iterations is more proper.
5. Unfair comparison to modern solvers. The pricing rules in most modern LP solvers are designed to take iteration as fast as possible. Generally, rules like the steepest pivot rule are not even the one-step greedy rule. Thus, comparing the number of iterations with them is not so fair.
6. Missing experiments on dual simplex and on OOD data. Generally, dual simplex is more preferred by LP solvers as the default LP approach. Thus, experiment on dual simplex is also critical. Experiments on OOD data is also critical to test the generalization ability.

[1] Li, Anqi, et al. "Rethinking Optimal Pivoting Paths of Simplex Method." arXiv preprint arXiv:2210.02945 (2022).

[2] Liu, Tianhao, et al. "Learning to Pivot as a Smart Expert." arXiv preprint arXiv:2308.08171 (2023).

[3] Suriyanarayana, Varun, et al. "DeepSimplex: Reinforcement Learning of Pivot Rules Improves the Efficiency of Simplex Algorithm in Solving Linear Programming Problems." (2019).

**Questions:**

What if using the optimal basis directly as the oracle? Intuitively, if we obtain the optimal basis, then they can serve as the oracle as they should be selected into the basis.

---

> ### Author Response · Authors · 2023-11-21
>
> W1: Thanks for the comment. To further clarify our motivation, we have included an additional paragraph in both the introduction and Section 2.2. Please review
>
> W2: This is a valid point. However, we tried to follow the strategy used in the relevant literature, e.g. [3] that uses TSP as test problems. Our primary goal in the computational part was to compare our RL simplex method with the mainstream commercial solvers based on the simplex method, such as Gurobi and SciPy. This type of comparison is itself something that not many studies in the existing literature have done. Within the scope of 50 cities, solving instances within the specified iteration limit has become exceedingly challenging.
>
> In the latest version of our paper, we have added a section for article reviews. We have also compared our algorithm with several other recent articles on machine learning and the simplex method published in recent years. We found that the test set we considered far exceeds the size of the largest test set used in other articles (over 10x), including those mentioned in this comment. Overall, we believe that the RLSimplex algorithm can address problems that can be solved by the simplex method. Most importantly, research in this field is still in its early stages, and we plan to delve deeper into this direction in the future.
>
> W3: Thanks for the comment. In the new version of the paper, we have added some related articles and conducted a comparative analysis of our algorithm with others that employ simplex and machine learning approaches. It is worth noting that due to the unavailability of their source code, our comparisons are based solely on the results presented in the respective papers.
> Regarding the strong branching issue, in general, this technique is primarily used to tackle MIP problems and has never been considered in the context of linear relaxation problems before. Furthermore, for large-scale problems, strong branching tends to impose significant computational demands. Additionally, it cannot rely on a rule similar to our algorithm's largest pivot rule to narrow down the selection space during the exploration process. This could be an interesting direction, but it requires further testing for comparison in the future.
>
> W4: During the learning process, we expect RL Simplex to explore a wide range of possibilities (i.e., different actions).  Therefore, we have incorporated UCB techniques to ensure a balance between exploration and the pursuit of high rewards, avoiding a singular focus on high rewards at the expense of neglecting other potential actions.  Additionally, if introducing a proportion into the rewards, we believe it is necessary to pre-calculate or estimate the required number of iterations for instances within that range.  However, this poses a significant cost, especially for large-scale problems.  In practical terms, such estimations are also challenging.  Taking our tested 50-city TSP problem as an example, the choice of the same numerical value but different variables as pivots can result in a difference of over 4850+ iterations. Additionally, experiments in paper [1] have also demonstrated the success of this straightforward reward design approach.
>
> W5: Based on the two articles provided by the reviewer, both studies [1][3] compare the steepest rule with other rules. From the results, it is evident that the steepest rule requires fewer iterations compared to other rules, such as our largest pivot rule. Furthermore, while we acknowledge that solver runtime is a crucial consideration, it is not explicitly stated here that solvers completely disregard iteration counts in their design, solely pursuing faster runtimes. Additionally, what we want to emphasize is that in other relevant literature[1-4], the comparison of solvers is typically based solely on the number of iterations. We understand that developers implement various computational improvements and different matrix decompositions based on the structure to reduce solution time and the number of iterations. We are not attempting to replace existing solvers in our paper; our focus is solely on the algorithmic aspect.
>
> W6: This is indeed an important direction. Due to time constraints, we plan to focus on the study of the Dual Simplex method in more detail in a different research. We had considered testing on some classic linear programming instance datasets, but our perspective aligns with that of [4], acknowledging that the existing data scope and scale are insufficient to meet our testing expectations compared to our standards.
>
> Q1: Thank you for your suggestion. Utilizing optimal solutions from previously solved TSP problems as feasible solutions/bases will significantly benefit the training process. Due to time constraints, we plan to incorporate this aspect into our future studies. The primary motivation for our current study is to propose and validate the feasibility of integrating reinforcement learning with the Simplex method.

---

### Official Review · Reviewer_tujU · 2023-10-29

**Soundness:** 2 fair
**Presentation:** 2 fair
**Contribution:** 3 good
**Rating:** 6
**Confidence:** 2

**Summary:**

This paper proposes a novel reinforcement learning (RL) based algorithm to select the pivot variables in simplex method for linear programming (LP). Numerical experiments demonstrate the effectiveness of the proposed RL simplex algorithm, outperforming established non-ML solvers in the Euclidean Traveling Salesman Problem (TSP).

**Strengths:**

1. The paper's idea of integrating RL in the simplex method for solving LP is new. It is also novel to incorperate the UCB method to balance exploration and exploitation while learning the optimal pivot rule.
2. In the experiments of the Euclidean TSP problem, the proposed RL simplex method outperforms existing non-ML LP solvers.

**Weaknesses:**

1. The RL approach section (Section 3.1) is not well organized and expressed. The methods are mostly descriptive, lacking rigorous mathematical statements. This makes it somehow hard to follow every detail, especially when the reader wants to reproduce the method for future research.
2. The method is only tested on the Euclidean TSP problem. It would be more convincing if more experiments on other LP problems can be conducted.

**Questions:**

1. Regarding the action space and reward function design, does that means whenever there exists a positive non-basic variable in $s_{t+1}$, then the reward received is $-kt$ *regardless* of the specific action $a_t$ chosen from possible largest non-basic coefficients?
2. There is recently a large body of literature on RL-based (mixed) integer linear programming algorithms, which is also extensively cited in this paper. However, it seems that only a little is discussed about the literature on solving standard LP problems assisted with ML methods, which is the focus of this work. Can you provide more about this line of research? Also, there is no comparison with existing methods for ML-based LP algorithms. How is the RL simplex method compare with other ML-based algorithms?

---

> ### Author Response · Authors · 2023-11-21
>
> W1: Thank you for your suggestion. We have provided more detailed descriptions for some definitions in Section 3.1, and made revisions in other sections to enhance the overall coherence of the article.
>
> W2:  Overall, we followed the approach used in the ML or RL-based simplex [1-4] and tested our method using standard TSP. We will continue to pursue research in this field and aim to expand the scope of the problem to other types of LP problems.
>
> Q1: Yes, because the simplex method requires verifying whether all non-basic variables are negative before proceeding to the next iteration. Because the algorithm has not yet satisfied the termination condition, the reward generated by the new iteration is -kt. One similar reward setting can be found in reference [1].
>
> Q2: Thanks for the comment. We have included the relevant literature on machine learning and simplex algorithm-related research in our latest edition. We intended to compare our work with other relevant approaches;  however, we could not get access to the open-source code of these algorithms, making it challenging to provide detailed comparisons in the experimental results. We have attempted to contact the authors of the relevant papers, but unfortunately, we have not received any response.
>
> Nevertheless, we can still conduct some preliminary comparisons by referencing dataset sizes or the use of similar modern solvers in other literature. For example, in the Simplex MCTS algorithm [1], the authors conducted a comparison with the SciPy solver as well. From Table 3 in the literature [1], it is evident that all problem ranges can be solved within 900 iterations. However, in our 50-city instances, utilizing the same SciPy Linprog Solver failed to resolve the problem within 5100 iterations. Therefore, we cannot ascertain the stability of Simplex MCTS on large-scale datasets.
>
> Another comparison is with the DeepRL Simplex, sharing a similar problem context, namely the TSP. From the test results [3] for the 5-city TSP problem, we both used Dantzig (Largest Private Rule) as a benchmark. Although the results suggest that DeepSimplex performs slightly better than ours in the 5-city TSP problem, it is essential to emphasize that DeepSimplex overlooks the impact of different city intervals on the simplex method, particularly the influence of Dantzig Largest Private Rule. This conclusion can be drawn from our Table 1 - City Range Analysis. We noticed an enhancement in the performance of the Dantzig Largest Private Rule with larger city intervals. However, it's important to note that the results presented in Table 2 are derived from a range of 1-5.

---

### Official Review · Reviewer_TGjj · 2023-10-30

**Soundness:** 2 fair
**Presentation:** 3 good
**Contribution:** 1 poor
**Rating:** 1
**Confidence:** 5

**Summary:**

A reinforcement learning approach for selecting pivot variables in the simplex algorithm is proposed. The algorithm is examined on the linear relaxation of the Euclidean traveling salesman problem (TSP). Note that this paper does not use deep learning; rather it uses (classic) Q learning to select pivot variables. The approach is tested on extremely small problems and, I emphasize, does not solve the real TSP, it only solves the linear relaxation.

**Strengths:**

The results are promising preliminary results that may lead to an interesting paper one day. I suppose the application of learning within the simplex algorithm is new, but I really question whether it makes any sense. Modern solvers are very fast and use simple rules for a good reason. This paper has a high hurdle to clear to be accepted.

**Weaknesses:**

The approach is very simple and tested on a single, extremely easy problem domain with tiny instances. The approach is simply not interesting unless it is applied to general LPs. Nobody needs a faster variable selection scheme for the TSP on 5 instances. Even for 50 instances, solving the problem is currently trivial in Concorde -- and then at least I get the optimal solution and not an optimal LP relaxation! The experimental analysis ignores the time required to solve instances, looking only at iterations. Thus, the time required for querying the Q-table is not included. And note that this is actually the interesting question: is the application of a "smart" pivot selection worth the time it takes to query the model?

**Questions:**

I have no questions.

---

> ### Author Response · Authors · 2023-11-21
>
> W1: It is indeed a valid point that solving instances with 50 cities in the context of TSP is not a large-scale problem in practice.  Our aim is to demonstrate the potential of RL in optimization.  We have followed the approach undertaken in the relevant literature ([1-4]) in which TSP has been used as the test problem.  These comparison articles also do not claim that they can beat Concorde, nor do we.   While other studies often do not compare their proposed methods with other solvers, we did compare the number of iterations required to find optimal solutions of LP relaxation using the established LP solvers and our proposed method.
>
> Furthermore, we included a TSP problem with 5 cities in the results to comprehensively record the performance of various methods under different data scales.  In addition, optimizing small-scale instances is essential, especially in the context of distributed linear programming problems.
>
> Finally, regarding the Q-table exploration issue you raised, querying from the Q-table takes run time, however, it contributes to reducing additional computations arising from the selection of different pivots.  In our latest version, specific experiments were conducted in Section 5 regarding this matter.  For more details, please refer to our article.  In summary, the additional time incurred by querying the Q-table is acceptable in terms of the overall computation time, especially when dealing with large-scale problems.  This proportion accounts for only 2% of the total computation time.  In comparison to non-learning algorithms, this can result in a reduction of over 3000 iterations.
>
>
> References:
>
> [1] Li, Anqi, et al. "Rethinking Optimal Pivoting Paths of Simplex Method." arXiv preprint arXiv:2210.02945 (2022).
>
> [2] Liu, Tianhao, et al. "Learning to Pivot as a Smart Expert." arXiv preprint arXiv:2308.08171 (2023).
>
> [3] Suriyanarayana, Varun, et al. "DeepSimplex: Reinforcement Learning of Pivot Rules Improves the Efficiency of Simplex Algorithm in Solving Linear Programming Problems." (2019).
>
> [4]Adham, Imran, Jesus De Loera, and Zhenyang Zhang. "(Machine) Learning to Improve the Empirical Performance of Discrete Algorithms." arXiv preprint arXiv:2109.14271 (2021).

---

### Official Review · Reviewer_95G6 · 2023-10-31

**Soundness:** 2 fair
**Presentation:** 1 poor
**Contribution:** 1 poor
**Rating:** 3
**Confidence:** 3

**Summary:**

The paper proposes to use reinforcement learning methods to help Simplex, a widely used mathematical technique for solving LP problems, to select variables during the pivot operation process. UCB algorithm is used to balance the exploration and exploitation during the RL process. The improved RL Simplex is used to solve the Euclidean Traveling Salesman Problem (TSP). The experiment results show that the method can reduce iteration requirements while maintaining optimal solutions for the Euclidean TSP instances.

**Strengths:**

This paper introduces a reinforcement learning approach that synergizes UCB and Q-learning to enhance the performance of the Simplex method to solve the LP problem. Additionally, it converts Traveling Salesman Problem (TSP) instances into linear programming problems, which are subsequently addressed by the refined method. The experimental findings highlight that for small TSP instances (comprising fewer than 50 cities), the proposed method significantly reduces the number of iterations when compared to baseline algorithms like Scipy and Gurobi, among others.

**Weaknesses:**

1.	All font sizes in Figure 1 are not uniform. Besides, what is the meaning of A, B and C in the Q-table.
2.	There may be some mistakes in the acting phase part. As the definition says, the parameter cij denotes the distance or cost between each pair of cities on the tour. But the definition of at is a set of largest cij corresponding to the variables of the objective function, which are not actions.
3.	The paper contains some abbreviations without full names, such as LP problem.
4.	The paper could be easier to follow, especially the description part of the key.
5.	The content marked in red in the experimental results part of the paper needs to be correct. For instance, in the row of 50 cities in Table 2, the solution result of HiGHS solver is better than that of RL Simplex, but the solution result of RL Simplex is incorrectly marked in red. Besides, Table 1 does not draw the optimal results.
6.	This paper mentioned that there has been previous work that used the RL method in Simplex, and also tried it on TSP. What is the difference between the method proposed in this paper and this method? Why not compare RL Simplex with this method?
7.	Some formulas are missing numbers, such as those in the Action Space part, UCB part, and Q-Value update part.

**Questions:**

1.	Please answer the questions posed in weakness.
2.	Experimental results show that this method can reduce iteration requirements. Still, whether the time consumed in each iteration is improved compared to the baseline, that is, whether the time to obtain the optimal solution is shorter than the original method.
3.	RL Simplex is particularly effective in improving the efficiency of solving linear programming problems in datasets with small sizes. However, a question arises regarding its performance when applied to larger datasets. Can you provide instances where traditional simplex methods fail to solve while RL Simplex successfully finds a solution? Such instances would serve as compelling evidence of RL Simplex's capabilities.
4.	As in the paper, if the solver meets a key not in the Q-table, it will always follow the largest coefficient rule. But as I understand it, most of the keys should be previously unencountered, especially in the process of doing different instances. Please give further instructions on how to apply the Q-table obtained on the training set to the test set? Besides, is it necessary to train a Q-table for different city-size instances? If it is needed, what is the generalization of this solver?

---

> ### Author Response · Authors · 2023-11-21
>
> W1: Thank you for your note. The updated Figure 1 has been incorporated into the new version of the paper. Regarding the elements A, b, c, these definitions are also explained in Section 2.1.
>
> W2: We have improved the explanation of actions in Section 3.1 of the article. We removed the corresponding expression but we provided clarifying descriptions.
>
> W 3: It has been corrected in the new version of the paper.
>
> W 4: We have provided further clarification on the key in Section 4.1, Learning and Acting Phase. We also streamlined the exposition to better reflect the description part of the key.
>
> W 5: We have now marked both the best-performing method and the second-best-performing method in Table 2.
>
> W 6: Please consult the recently added literature review (Section 1) in our paper, as well as the explanation provided in the Unified Responses for Committee and Reviewers above.
>
> W7: It has been corrected in the new version of the paper.
>
> Q2: Thank you for your advice. Regarding the issue of comparison methods, please refer to our recently released general responses above for more information
>
> Q3: From our current results and literature review, it appears that the TSP with 50 cities already poses significant computational complexity on solvers. In testing for large instances, we set a limit of 5100 iterations, far exceeding the tolerance for iteration counts observed by peers[1-4]. However, without such limits, existing solvers, including the SciPy solver and Gurobi, can reach iteration counts exceeding 8000 without convergence in some instances. It is crucial to emphasize that the HiGHS solver, unlike our other testing methods, does not solely rely on the simplex method to solve LP problems.  Hence, it exhibits advantages in handling large datasets.  We chose to include the HiGHS solver in the comparison to understand the extent of the difference between our algorithm and a solver specifically designed for large-scale data. This comparative approach is being applied for the first time in related research articles with a similar focus.
>
> Q4: We suggest training on different city scales. This is because the number of variables and constraints vary significantly with different city sizes in the TSPs, as indicated in the leftmost column of Table 2. We can train TSPs with different city sizes together and place them in a trained Q-table. However, this would increase the time required for key matching during testing.
> Additionally, regarding how to apply the trained Q-table to new instances, traditional Q-learning learns from past experiences based on state. Still, in our algorithm, the definition of state is slightly different—please refer to Section 3.1. Therefore, we need to extract the Cij part of each iteration as a key to help us find past learning experiences from the trained Q-table.
> It's worth noting that the simplex method is not a one-shot algorithm; we can consider each iteration as solving a new LP problem. Therefore, new instances can also gain experience from the solving process of previous instances. While training for large-scale TSP problems may require a significant number of iterations (exploring different actions), it proves beneficial for testing new instances.

---

> > ### Comment · Reviewer_95G6 · 2023-11-23
> >
> > Thanks for your response. After reading the responses, some of the concerns have been addressed and the revised paper is improved.
> >
> > However, although the paper demonstrates that the effectiveness of the simplex can be improved by incorporating reinforcement learning, the RL Simplex used for solving the TSP problem can still only solve extremely easy instances. I don't agree with the comparison approach that only considers the iterations. Excluding factors such as fairness, based on the Figure 2, the time required for querying the Q-table still has a high percentage of total time.
> >
> > I have also gone through other reviews, and tend to keep my current rating.

---

### Author Response · Authors · 2023-11-21
**Unified Responses for Committee and Reviewers**

1. Motivation:

First, we aim to enhance traditional algorithms through a machine learning framework for more effective problem-solving.  The core idea behind our approach is intuitively understandable but often overlooked.  Existing literature in the field tends to focus on guiding rules in different instances[3-4] or utilizing machine learning to propose new rules[1-2].  In contrast, our contribution lies in improving classical rules through machine learning, which has proven effective in our results.  Second, we chose the TSP as the primary optimization for testing the approach because it is a well-known problem while also allowing us to demonstrate the potential of the approach by increasing the complexity by increasing the number of cities.  However, it's crucial to note that our intention wasn't to solve the TSP problem itself, as highlighted in [5]  Even though machine learning can be beneficial for solving TSPs, it still doesn’t yet address the complexity found in real-world problems.  Our purpose in doing so is to prove and showcase the feasibility and potential of the core idea of our approach.

2. Comparison with Peers:

We have actively reached out to the authors of the relevant (comparison) papers to request access to their code, but we have not received any response. Consequently, we are unable to provide specific comparative results in the experimental section of our paper. To address this limitation, we have incorporated a literature review section in the latest version of the paper. This section focuses on reviewing several articles that share similar research directions and contributions. Despite this limitation, we were able to make approximate comparisons with the results presented in these comparison articles. For instance, in the case of the 5-cities TSP problem tested in [3], the results of DeepSimplex are slightly better than ours. However, we want to emphasize that DeepSimplex did not fully consider the impact of distance variations on the model during testing. As shown in Table 1 in our paper, different distances between cities can affect traditional pivot rules, especially the largest pivot rule (Danzig rule). To mitigate this impact on our results, all the results presented in Table 2 are based on the minimum distance changes between cities. Furthermore, from the results shown in Table 2 of the DeepSimplex paper, we found that pursuing high-reward actions can lead to overlooking the potential gains from other actions. Therefore, in this process, we propose to use the UCB concept in Q-learning to help address this issue. In addition to comparisons related to these algorithms, we also noted that the test scale we provide is the largest among these articles [1-4] (more than 10x larger), which can be compared using the dimensions of the constraint matrices and the number of iterations required by the same solver as provided in the results section of these articles.

3. Size of the Test Set:

We have received some comments regarding the size of our test set. As explained earlier, we chose the TSP as a test bed to validate the proposed approach. Additionally, we are mindful of the scalability of the algorithm, and therefore, we selected TSP instances with 50 cities as one of our test cases. This not only surpasses the complexity of the test environments used in similar papers but also, at this scale, we have observed that modern solvers are unable to find the optimal solution within the specified iteration limit. This is a contribution highlighted for the first time within the related research.

4. Comparative Methodology:

Regarding the comparison of computational time, we observed that the majority of relevant papers [1][3-4] did not utilize runtime as a comparison criterion. The solvers we employed also output the iteration count as one of the output results, thus we did not consider run time in the comparison results.   The iteration count is a more fair comparison than run time across methods as it is not dependent on efficient software implementations or hardware upgrades.  The iteration count can more intuitively reflect the logic of the algorithm, unaffected by computational power.  Additionally, unlike other articles, we found that some solvers, such as SciPy and even Gurobi, were unable to provide the optimal solution within the specified computational environment in our test set of 50 cities.  Therefore, it is unfeasible to compare them with other viable solution methods in terms of run time. In practice, besides run time, the success rate of finding the optimal solution is also a crucial metric.
In addition, we tested the time proportions of key functionalities within the algorithm across different city ranges, as detailed in Section 5. In summary, steps that might be perceived to incur additional computational costs, such as Q-table queries, constitute only 2% of the total time in large-scale instances, which is deemed acceptable.

---

> ### Author Response · Authors · 2023-11-21
> **References**
>
> [1] Li, Anqi, et al. "Rethinking Optimal Pivoting Paths of Simplex Method." arXiv preprint arXiv:2210.02945 (2022).
>
> [2] Liu, Tianhao, et al. "Learning to Pivot as a Smart Expert." arXiv preprint arXiv:2308.08171 (2023).
>
> [3] Suriyanarayana, Varun, et al. "DeepSimplex: Reinforcement Learning of Pivot Rules Improves the Efficiency of Simplex Algorithm in Solving Linear Programming Problems." (2019).
>
> [4]Adham, Imran, Jesus De Loera, and Zhenyang Zhang. "(Machine) Learning to Improve the Empirical Performance of Discrete Algorithms." arXiv preprint arXiv:2109.14271 (2021).
>
> [5]Bengio, Yoshua, Andrea Lodi, and Antoine Prouvost. "Machine learning for combinatorial optimization: a methodological tour d’horizon." European Journal of Operational Research 290.2 (2021): 405-421.

---

### Meta-Review · Area_Chair_t2Pc · 2023-12-09

**Metareview:**

Paper introduced an approach, namely RL Simplex, to accelerate the solver of Euclidean Traveling Salesman Problem (TSP) with claimed superiority over commercial packages, e.g., Gurobi and SciPy. While the paper presents interesting ideas, reviewer generally expressed concerns on problem motivations, the limited size of TSP problems that the approach solves, and lacking large-scale experiments and SOTA baselines.

Paper needs to address the above issues to reach the threshold of acceptance.

**Justification For Why Not Higher Score:**

Concerns raised by reviewers needed to be addressed before paper can be reconsidered for acceptance.

**Justification For Why Not Lower Score:**

N/A

---

### Decision · Program_Chairs · 2024-01-16

Reject